# Impact of IoT System Imperfections and Passenger Errors on Cruise Ship Evacuation Delay

**DOI:** 10.3390/s24061850

**Published:** 2024-03-13

**Authors:** Yuting Ma, Erol Gelenbe, Kezhong Liu

**Affiliations:** 1Instytut Informatyki Teoretycznej i Stosowanej Polskiej Akademii Nauk, 44-100 Gliwice, Poland; 2School of Navigation, Wuhan University of Technology, Wuhan 430063, China; kzliu@whut.edu.cn; 3Université Côte d’Azur, CNRS, I3S, 06100 Nice, France; 4Department of Computer Engineering, Yaşar University, Bornova, 35500 Izmir, Turkey

**Keywords:** passenger evacuation, naval vessels, cruise ships, emergency systems, information technology support, processing delays, congestion in communication systems

## Abstract

Cruise ships and other naval vessels include automated Internet of Things (IoT)-based evacuation systems for the passengers and crew to assist them in case of emergencies and accidents. The technical challenges of assisting passengers and crew to safety during emergencies include various aspects such as sensor failures, imperfections in the sound or display systems that are used to direct evacuees, the timely selection of optimum evacuation routes for the evacuees, as well as computation and communication delays that may occur in the IoT infrastructure due to intense activities during an emergency. In addition, during an emergency, the evacuees may be confused or in a panic, and may make mistakes in following the directions offered by the evacuation system. Therefore, the purpose of this work is to analyze the effect of two important aspects that can have an adverse effect on the passengers’ evacuation time, namely (a) the computer processing and communication delays, and (b) the errors that may be made by the evacuees in following instructions. The approach we take uses simulation with a representative existing cruise ship model, which dynamically computes the best exit paths for each passenger, with a deadline-driven Adaptive Navigation Strategy (ANS). Our simulation results reveal that delays in the evacuees’ reception of instructions can significantly increase the total time needed for passenger evacuation. In contrast, we observe that passenger behavior errors also affect the evacuation duration, but with less effect on the total time needed to evacuate passengers. These findings demonstrate the importance of the design of passenger evacuation systems in a way that takes into account all realistic features of the ship’s indoor evacuation environment, including the importance of having high-performance data processing and communication systems that will not result in congestion and communication delays.

## 1. Introduction

In case of emergencies, evacuation methods are critical to managing people and vehicles in a manner that guarantees their safety [1,2]. These methods rely on the sensing, communication, and signaling technologies that are required to (a) know where people and vehicles are located and (b) how one can communicate with them or inform them about ongoing conditions so as to (c) direct them along effective pathways towards safety.

Research in this area includes the use of sensing and communication technologies [3], crowd monitoring [4], hazard modeling and prediction [5], evacuation simulation and evacuation path planning [6]. In particular, offline simulation of evacuation schemes aids the design and comparison of the sensing and communication technologies, and of the algorithms [7] that improve or optimize the performance and robustness of evacuation strategies.

Thus, emergency management simulation research addresses simulations that represent the movement of people who congregate in sports arenas, touristic sites, and other leisure venues [8,9], as well as in large ships [10], in the presence of unusual and extreme conditions such as the breakdown of some facilities or adversarial situations such as fire or panic.

With the increased popularity of the cruise ship industry worldwide, more attention has been paid to passenger safety in maritime transportation [11]. Although advanced accident prevention systems are deployed in modern passenger ships, maritime transport accidents unfortunately occur, such as the Sewol Ferry accident in South Korea that caused 304 casualties on 16 April 2014 [12], the Chinese Eastern Star accident that caused 42 casualties on 1 June 2015 [13], and the collision of two passenger ships in the Padma River in Bangladesh, resulting in 26 deaths on 3 May 2021 [14]. Therefore, effective evacuation methods for passengers on cruise or passenger ships are of great importance [15], and significant research is recently being conducted in this area [16].

Since such accidents or emergencies cannot be artificially created or reproduced or easily observed for data analysis when they occur, the simulation of human evacuation on ships has become a key tool in the design of both civilian and military naval vessels [17,18,19].

In particular, the IMO (International Maritime Organization) has published circulars regarding the simulation of naval vessel evacuation both for passenger and roll-on-roll-off ships. In 2016, it also issued approvals for guidelines regarding the simulation of evacuation both for new and existing passenger ships [20]. However, the actual technology that is used in supporting passenger evacuation, such as the computers, communication systems, visible panels, loudspeaker announcements, and other sensors and actuators, can themselves have some imperfections that are exacerbated during an emergency. Furthermore, passengers can also be panicked and confused during such events. Thus, this paper focuses on the effect—*on the time taken by passengers to exit the ship*—of:Computation and communication delays in the sensing, communication and computer processing technologies in the vessel, including the processing delays in the computers that provide instructions to the evacuees [21,22];Errors that the passengers may make in following instructions. Such errors can be caused by panic, human error or the lack of visibility of direction panels, and the difficulty of understanding loudspeaker announcements due to noise.

Our study uses a simulation framework to evaluate the outcome of these factors on the time needed by passengers to reach a safe exit, based on multiple simulation runs using the AnyLogic simulator [23] described in Section 2. The present research does not consider some other features that can impact the speed and safety of human beings during emergencies [7,24], including the time needed to traverse passages and staircases as a function of the health and age of different passengers. However, our simulations include the effect of the changes in the tilt of the ship which will affect the traversal times.

While AnyLogic is used as part of our simulation tool, we do not claim or suggest that this tool should be used for real-time decision making during an emergency. We expect that the software used to manage evacuations during the emergency would include a fast discrete-event purpose-built simulator which updates—in real-time—the dynamic spread of hazards that are reported via sensors installed in the cruise ship, or in any other built structures such as buildings, campuses or cities, which require emergency evacuation.

The remaining sections of this paper are structured as follows: The related research about evacuation in passenger ships and built structures is reviewed in Section 1. A detailed description of the ship emergency evacuation simulator is provided in Section 2, while the simulation results regarding the impact of delay and lost messages, which result from communication system congestion, are given in Section 3. Then, Section 4 presents the possible impact of uncertainties in human behavior. In Section 5, we discuss potential extensions and open aspects of our work and present our conclusions.

### Related Work

There has been extensive work on using the IoT to support emergency management [25], and advanced systems such as Unmanned Aerial Vehicles (UAVs) have also been suggested as a means for observation and sensing [26] during emergencies. While the current study does not assume that evacuees were identified and tracked during the evacuation, there has been work on the issue of localization of specific groups, such as elderly people who may have greater difficulty in reaching exits safely [27].

While the use of path optimization algorithms in emergency evacuation is quite common [28] to minimize evacuation delay or minimize the risk to evacuees using prior 2D/3D knowledge [29,30], the decisions regarding the choice of paths do not always include the effects of unexpected or new hazards that may be encountered. The Expected Number of Oscillations (ENO) [31] has been used to quantify the dynamic changes due to spreading hazards, so that a small ENO indicates the paths do not change often under the effect of possible dangers and are therefore to be preferred. Indeed, methods that try to predict the optimum paths may not result in the safest paths, and the information available prior to an emergency may not match the realistic developments that may occur as an emergency unfolds. Therefore, recent studies do not require that all possible hazards be known for optimization purposes and tend to lead to results that have more value in practice [7,32].

Methods such as Social Potential Fields (SPF) [33] and local neighborhood techniques [34,35] with possible partial reversal can also help evacuees avoid entering hazardous areas. However, most studies do not consider the need to provide directions that assure that the “time needed to reach the exit” for each evacuee remains under a required specified bound determined by the characteristics of the worst-case dynamic characteristics of the ship.

Thus, this paper uses the Adaptive Navigation Strategy (ANS) evacuation algorithm, an extension of Rapid Routing with Guaranteed Delay Bounds [36], which was previously named ANT in [37], creating potential confusion with other well-known “ant colony” optimization techniques. ANS incorporates a guaranteed exit deadline bound, for each evacuee in each location. When dealing with ships, this deadline bound can be obtained, as already indicated above, from the recommendations of the IMO. We note that various organizations in various countries and regions have made corresponding safety recommendations for office buildings, apartment buildings, factories, university campuses, etc. Note that **all the work cited in this paper** is summarized in Table 1.

## 2. The Simulation Framework

We now describe the simulation framework that we use to evaluate the effects of both the computer and communication technology delay in providing routing instructions to evacuees, and the uncertainties related to the evacuees’ behavior, within an event-driven simulation framework. This framework contains two parts:The first part is the simulation software AnyLogic 8.8-8.8.2: Please state the version number of the software. I have added the version number. in which the layout of the physical where the evacuation occurs is incorporated. The “pedestrian” software library of AnyLogic adopts the Social Potential Field model to determine the direction of the movement of each evacuee.Secondly, we add a path-planning module written in Python that computes the evacuation direction for evacuees based on ANS, which is described below. This module transfers the computed instructions to the AnyLogic simulation software at each simulation step when the movement instructions need to be updated.ANS is implemented in our simulator to move evacuees along the path with minimum delay, avoiding the harmful effects caused by dynamic hazards. ANS assumes knowledge about the propagation of hazards (velocity and direction), and the average and maximum delay across each edge in the paths.As hazards progress in the simulation, ANS calculates the direction each evacuee should take to avoid hazards. In a real evacuation, this direction should be computed and then communicated to each evacuee via a wired or wireless network. In previous work, the possible delays of this communication were not taken into account.However, wireless or wired networks and computational servers that are used for decision making are likely to experience congestion, especially in emergency situations when decisions and communications are frequently updated and many messages are sent to evacuees and to the staff in the ship. This congestion can cause delays in updates regarding the navigation direction, and transmitting network packets and hence messages can be lost, and decisions may lead to errors due to the arrival of delayed instructions or facts, used by decision algorithms, that have been modified by events [43,44]. While most prior work neglects these effects, the present paper specifically evaluates their effect on the time required for the evacuation.Also, the passengers being evacuated may themselves be unable to follow the instructions they receive due to noise, panic, or misunderstanding.Thus, these delays and possible errors due to Information and Communication Technology (ICT), including network packet losses, as well as the possible effects of panic or misunderstandings by the evacuees, will be simulated and evaluated in this paper.

### 2.1. The Supporting IoT System

The IoT could, in principle, use the personal identification of each passenger via “smart badges”, “RFID” (radio-frequency identity), or communicating “wristwatches”. However, such devices requiring radio-frequency communications can be unreliable within naval vessels. Indeed, large ships typically have high steel content with possible local Faraday cage effects, as well as the refraction and reflection of high-frequency radio waves. In addition, the numerous electric motors (e.g., for ventilation and lifts) and ongoing automatic switching of equipment (e.g., on–off of air-conditioning equipment, multiple refrigerators, fans), as well as multi-path reflections, can lead to substantial radio-frequency interference. Other means of sophisticated local communication, such as high-frequency ultrasound or modulated light signals, may potentially be used, but in this paper, we assume that individual identification of the location of each passenger is not available.

The ship that we are simulating will have an ICT infrastructure that includes a **high-speed local area network (LAN)** equipped with switches and routers that conveys packet-based data communications between all the sensors and actuators together with WiFi hubs as needed. The LAN is connected to a **Data Center (DC)**, that is used to compute the emergency instructions for the evacuees, as well as for storing and processing other information related to the passenger ship. The LAN topology, as well as the DC itself, will be designed in advance for appropriate redundancy and reliability to adequately face emergency situations.

Infrared sensors connected to the ICT infrastructure may be placed throughout the vessel (in cabins, corridors, etc.) to detect human presence without identifying the individual. Of course, such sensors can also detect hazards such as high temperatures caused by fire, and electrical short circuits. There will also be a variety of temperature and smoke sensors. Passage locations and common areas (e.g., restaurants, decks, bars, and lounges) are typically equipped with video cameras to estimate the number of people present, and also for the purpose of security.

In addition, throughout the ship and connected to the ICT infrastructure via the LAN, there will be various **Emergency Direction Providers (EDPs)**. These will activate in an emergency and will have red/white signs throughout the locations of the ship, which flash and provide directions to the passengers, with instructions such as “Exit Here”, “Turn Right”, “Turn Left”, “Go Straight”, “Wait Here”, etc., to provide directions to help evacuate the passengers. These instructions will be computed by the DC and sent over the LAN to the EDPs.

In case of an emergency, the DC runs the ANS evacuation algorithm to determine for the passengers, the **Evacuation Movement Recommendations (EMRs)**. Thus, the simulations in the present paper evaluate the effectiveness of the EMRs in the presence of the following:Delays in the reception of the EMRs at the locations of the EDPs throughout the vessel. These delays can be caused by LAN delays and congestion, and DC delay and congestion during an emergency.Errors made by evacuees in following instructions from the EDPs during an emergency evacuation due to confusion and panic.

### 2.2. System Parameters for the Simulation

The indoor environment we simulate is the second, third, and fourth floors of the Yangtze Gold 7 Cruise Ship, as shown in Figure 1 and Figure 2, with 346 nodes from which evacuees originate or pass through, including connection points between corridors or rooms through which evacuees may pass, and a single exit node for the evacuation. There are also 600 passageway segments and 5 staircase segments at the edges of the graph that connect these nodes.

The simulation uses the worst-case evacuee movement speed to estimate the time required for evacuees to cross each segment, which is 0.067 m/s. The average traversal time for each segment is estimated from the passengers’ average speed of movement calculated based on the average speed at which an evacuee walks when the ship is horizontal is taken to be 0.67 m/s. In addition, the total time available to a passenger to evacuate the ship when they receive the “evacuate” message can be estimated as follows:(1)$$T_{D}=T_{S}-T_{A}-T_{EL},$$
where $T_{S}$ is the ship’s survival duration (e.g., until it capsizes), $T_{A}$ is the delay between the start of the emergency until the “evacuate” message is received by the evacuees, and $T_{EL}$ is the time needed to embark on a lifeboat and for the lifeboat. The MSC (Maritime Safety Committee) [20] has estimated that $T_{S}$ = 60 min, $T_{A}$ = 5 min, $T_{EL}$ = 25 min; thus, $T_{D}$ = 30 min.

### 2.3. Layout of the Simulation Framework

The simulation layout for the Yangtze Gold 7 cruise ship’s three passenger floors in Figure 2, shows the “dots” (i.e., nodes) which are locations where passengers may be staying or meeting (e.g., cabins, lobby, and restaurant), while the edges show the passageways, stairs, or corridors. The simulation also allows the inclusion of the effect of the inclination angle when the ship is damaged, which can change at regular intervals. This can affect the average traversal time encountered across each individual corridor or staircase which changes (shorter or longer time) with the inclination.

Each simulation is initialized by locating the evacuees (passengers) at random over the nodes, and each simulation is repeated over 100 rounds where the initial locations each time are randomized in the same way.

## 3. Impact of Delays in Computing and Communications on Passenger Evacuation

The ANS provides each evacuee, at each of the nodes of all the evacuation paths in the ship, an estimate of the next-step hazard-free node that the evacuee should enter to head toward the exit, based on the estimated total minimum delay to the exit from its current location. Since conditions may rapidly change with time during an evacuation, the computer and communication system that computes these directions and forwards the decisions to the nodes on each path may be congested during an evacuation. Thus, the resulting messages to evacuees may be delayed or lost.

Therefore, in this section, we analyze the impact of these possible delays, which are caused by performance imperfections and congestion of the underlying Information Technology System (ITS). To this effect, we **define the “information lag” (*****IL*****)** which refers to any generic node in the ship evacuation topology. IL=0 means that each node in the evacuation topology of nodes and paths has received, from the ITS system, the exact direction recommendation which is based on the current true location of the evacuees at each node.

On the other hand, IL=1 means that each node provides information to the evacuees regarding the next move they should make based on the location of the evacuees just prior to the current arrival of evacuees to their current node. Thus, IL=1 means that the computation and transmission of the information are delayed by one step.

We also define the **“probability of delay” (*****PoD*****)**, which indicates whether the information lag is IL=1 with probability PoD or IL=1 with probability 1−PoD. PoD is a value that is probabilistically attributed to each node, since the delay may differ from node to node due to the communication system delays.

In the rest of this section, we evaluate the effect of PoD on the evacuation system’s performance. All simulation results that are shown are obtained by drawing the probability PoD separately for each of the nodes and for 100 independent simulations under the same initial conditions. The figures that are shown for each simulation also show the 95% confidence intervals.

### Evaluation of the Average Evacuation Time

The first evaluation is conducted to determine the **average evacuation time in seconds, from all of the 346 nodes**, relative to the ideal case with IL=0.

Figure 3a plots the average evacuation time from all 346 starting nodes until the exit as a function of PoD. Figure 3b shows the performance ratio of average evacuation time for different values of PoD to the average for the ideal case of PoD=0. Averages are taken over all nodes and based on 100 distinct independent simulations. We also show the standard deviation (black bars) for the evacuation time from all 346 nodes. These curves show clearly that as PoD increases to 0.5, the increase in average evacuation delay is quasi-linear, but for higher values, the increase continues but more slowly. When PoD=1 for all nodes, the average evacuation time over all nodes is 50% higher than PoD=0 (where the ITS provides up-to-date information to all nodes).

In addition, we also implement a group of simulations to evaluate the **average evacuation time, in seconds, for passengers located in cabins**. Figure 4a shows the effect of the probabilities PoD on the evacuation time of passengers that start from cabins. The performance ratio of the average evacuation time from cabins relative to the case PoD=0 is given in Figure 4b, showing that it increases with PoD.

We also evaluate the **average evacuation time for passengers that are initially located in the restaurant**. Figure 5a presents the results for passengers who started the evacuation from the restaurant. It appears that the information processing delay has a smaller influence in this case, as compared to the evacuation time of passengers from the cabins when the delay probability PoD is low, such as PoD=0.1 and PoD=0.2.

Finally, we perform a group of simulations in which half of the passengers are initially placed at random in cabins while the other half start from the restaurant, and we vary the total number of passengers. Note we assume that there are at most two passengers per cabin.

As shown in Figure 6a, the evacuation time grows with the number of passengers. This is mainly attributed to the increased waiting time resulting from the more serious congestion due to a large scale of passengers being evacuated. The evacuation time also increases with PoD, regardless of the number of passengers.

Figure 6b shows the change in the ratio of average evacuation time as a function of the number of passengers and of PoD. The performance ratio is always larger than 1, which indicates that as PoD grows, then the evacuation also duration grows, and the growth rate is higher for larger PoD.

## 4. The Effect of Uncertain Passenger Behaviour

During emergencies, the evacuees may panic or find it hard to read or hear instructions for evacuation. We model this by the choice of evacuees to miss the correct direction and take another one at random.

Therefore, in this section, we investigate the impact of the non-compliance with the evacuation suggestion. To this effect, we define the “**probability of error**” denoted as PoE, which indicates whether a passenger does not obey the provided evacuation direction with probability PoE or moves along the provided direction with probability 1−PoE. PoE is a value that is a probability that is chosen itself at random and attributed to each passenger, since the behavior may differ from passenger to passenger.

### The Average Evacuation Time

First, we conduct simulations to measure the evacuation time, from all of the 346 nodes to the exit under various values of PoE. Figure 7 presents the ratio of average evacuation times relative to the case PoE=0, where passengers fully comply with the ITS recommendation. Clearly, all the values of the performance ratio exceed 1, which shows that the behavior uncertainty of passengers prolongs the average evacuation time in seconds. We can observe that with the increase in the probability with which passengers ignore the evacuation guidance, the performance ratio in average evacuation time in seconds also increases. However, compared to PoD, the uncertain behavior of passengers has a less pronounced effect on the average evacuation time in seconds.

We also carry out a set of simulations to evaluate the impact of behavior uncertainty on the evacuation time of passengers who are initially located at random at nodes representing passenger cabins. Figure 8 plots the average evacuation time and the performance ratio compared to the case PoE=0. The standard deviation is also shown. It can be observed that the average evacuation time increases with the probability of behavior uncertainty. When PoE=0.5, the average evacuation time increases by almost 40% relative to PoE=0. Furthermore, it is worth noticing that the passengers in cabins are more significantly affected by their uncertain behavior compared to the results in Figure 7.

The average evacuation time in seconds for passengers in the restaurant is also measured as shown in Figure 9. We can see that failing to exit according to the instructions impairs the evacuation scheme and lengthens the evacuation of passengers from the restaurant. Again, it appears that this uncertain behavior has a milder effect on the average evacuation time for passengers in the restaurant compared to passengers leaving from the cabins.

We also measure the average evacuation time for different numbers of passengers with different probabilities of not following the evacuation directions. Here, half of the passengers start from the restaurant while the other half start from their cabins. Figure 10 plots the average evacuation time with the 95% confidence interval for different numbers of passengers with different probabilities of uncertain behavior. We can see that the average evacuation time increases with the probability of uncertain behavior. Nevertheless, we observe that behavior uncertainty has a much smaller effect on the average evacuation time, particularly if the passenger does not exceed 300. Again, behavior uncertainty has a much lower impact than PoD, showing that the ANS method is more resilient to uncertainties in evacuee behavior than to delays in the information technology system used for directing evacuees.

## 5. Conclusions and Future Work

Large transportation systems such as trains, passenger ships and aircraft require reliable and effective emergency evacuation to ensure the safety and passengers. Thus, much research has addressed the design of technology-assisted means to direct passengers effectively during emergencies, with algorithms and methods that maximize fast and optimized means for evacuation [39]. Thus, the present work addresses some of the key side effects of technology-assisted emergency evacuation that includes extensive simulations of a realistic and existing ship scenario.

Assuming that ongoing situational information is gathered by sensor networks [40,41,42] and then processed to provide optimum decisions and communicated to the evacuees as they progress towards the designated exits, our work has included the effects of imperfections in the technology support such as communication and processing delays for instructions to arrive at the evacuees, as well as the effects of mistakes that evacuees themselves may make during an evacuation.

We see that delayed decisions which are forwarded to all end evacuees as they pass through pre-determined “nodes” that guide them towards safety will create substantial evacuation delays for the evacuees. Similarly, we have observed that mistakes that are made by passengers due to panic or misunderstanding of the instructions also will increase the evacuation delays, but to a lesser extent as compared to delays due to the technology.

Thus, further research is needed to support evacuee safety in such complex and dynamic environments with many rapidly changing effects that may overwhelm both the ICT-based emergency management system [38] and the ability of evacuees to follow instructions. We suggest that future research could include these delaying factors in advance and design novel decentralized emergency navigation systems that pre-locate advisory data and related computational means in key system intermediate locations, combining centralized decisions with local decision making individual evacuee decision aids and hand-held mobile devices [35].

## Figures and Tables

**Figure 1 sensors-24-01850-f001:**
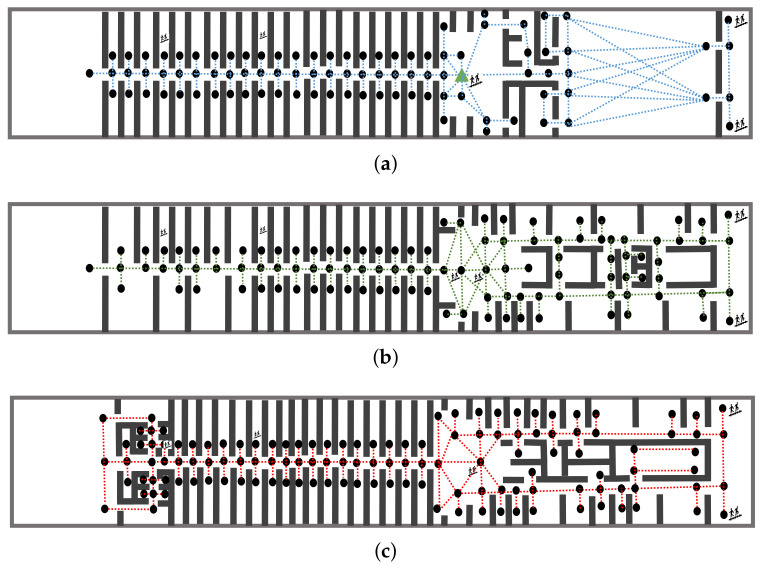
Schematic description of the layout for the Yangtze Gold 7 Cruise ship over three passenger floors (second, third, and fourth). This layout is used for simulating all the effects studied in this paper, including the delays in communicating the guidance information to the evacuees, and the possible uncertainty in the behavior of the evacuees. Here, (**a**–**c**) show the layout of the physical space of the second, third, and fourth floors.

**Figure 2 sensors-24-01850-f002:**
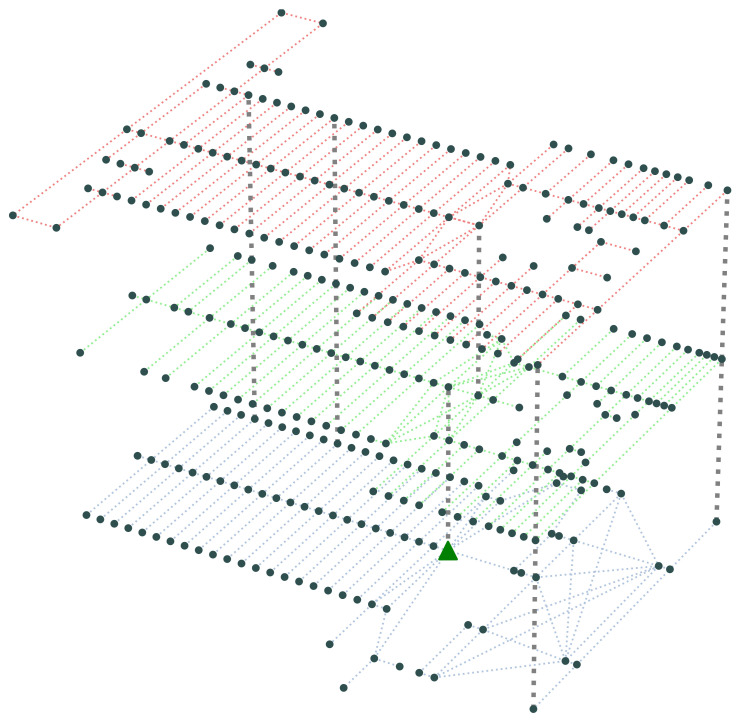
The evacuation graph representation of the physical space, showing the nodes where routing decisions are communicated to the evacuees via the EMRs.

**Figure 3 sensors-24-01850-f003:**
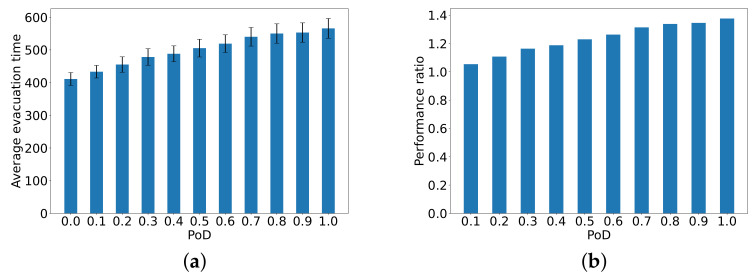
The average evacuation time in seconds from all 346 nodes to the exit (**a**) as a function of PoD (x-axis), and (**b**) the performance ratio of the average evacuation times as compared to the ideal case of PoD=0. Averages are over all nodes for 100 distinct independent simulations, with the standard deviation (the black bars) for the evacuation time.

**Figure 4 sensors-24-01850-f004:**
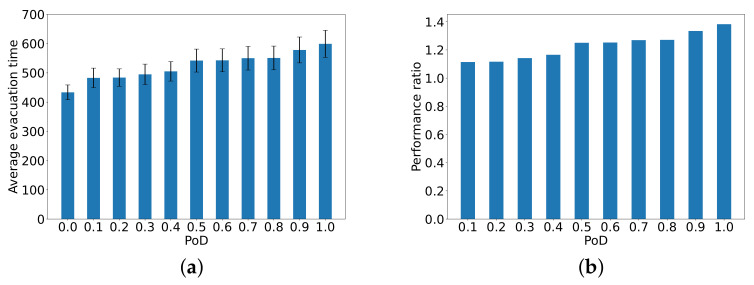
The average evacuation time from the cabins to the exit given in seconds (**a**) as a function of PoD (x-axis), and (**b**) the performance ratio of the average evacuation time as compared to the ideal case of PoD=0. Averages are taken over all passengers starting from cabins for 100 distinct independent simulations. The standard deviation (the black bars) of the evacuation time is also shown.

**Figure 5 sensors-24-01850-f005:**
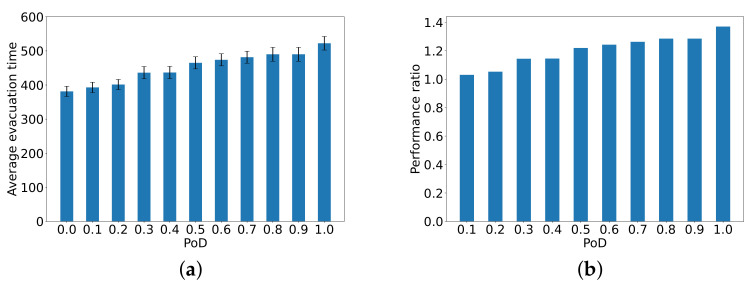
The average evacuation time in seconds for passengers that are initially located in the restaurant (**a**) as a function of PoD (x-axis), and (**b**) the resulting performance ratio as compared to the ideal case of PoD=0. Averages are taken over all passengers who were initially in the restaurant for 100 distinct independent simulations. The standard deviation of their evacuation time is shown with the black bars.

**Figure 6 sensors-24-01850-f006:**
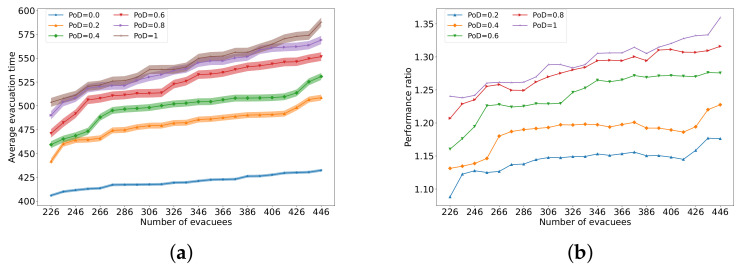
The average evacuation time in seconds, with a 95% confidence interval, taken by different numbers of passengers where half of them originate in the cabins while the other half start from the restaurant (**a**), and (**b**) the performance ratio of the average evacuation time as compared to the ideal case with PoD=0.

**Figure 7 sensors-24-01850-f007:**
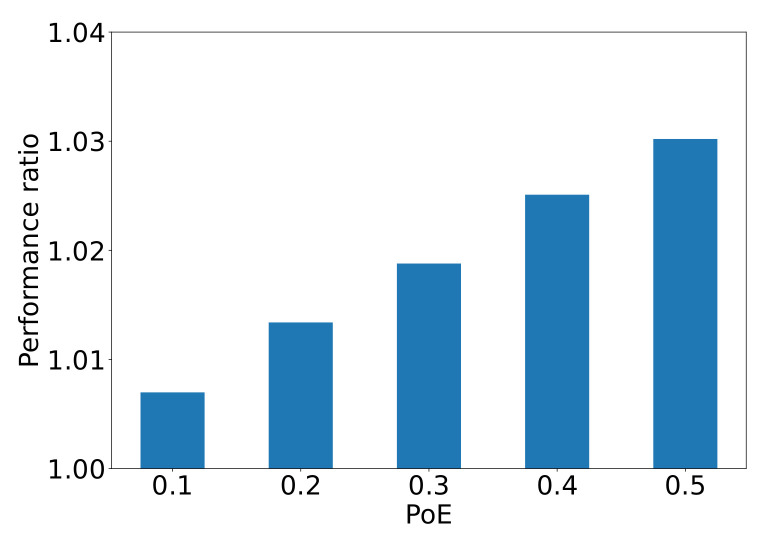
The performance ratio in average evacuation time in seconds from the 346 nodes to the exit compared to the ideal case PoE=0.

**Figure 8 sensors-24-01850-f008:**
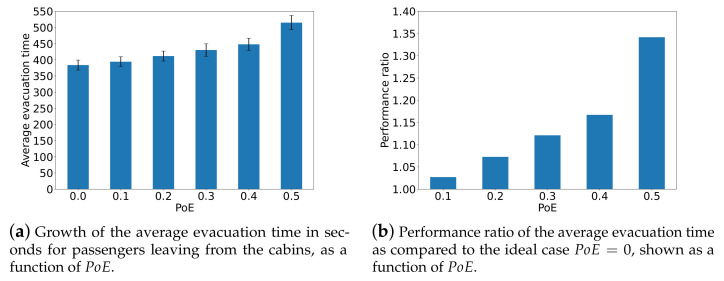
The average evacuation time in seconds taken by passengers in cabins and the performance ratio of the average evacuation time compared to the ideal case PoE=0.

**Figure 9 sensors-24-01850-f009:**
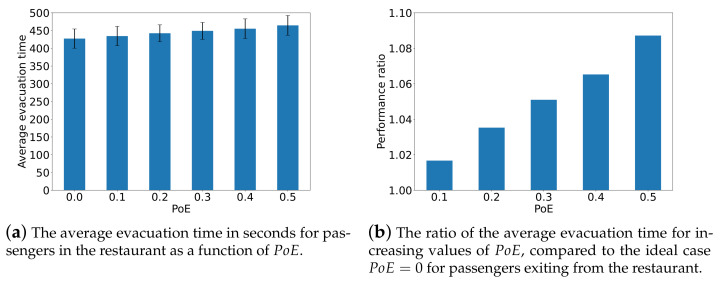
The average evacuation time in seconds for passengers in the restaurant and the performance ratio in average evacuation time compared to the ideal case PoE=0.

**Figure 10 sensors-24-01850-f010:**
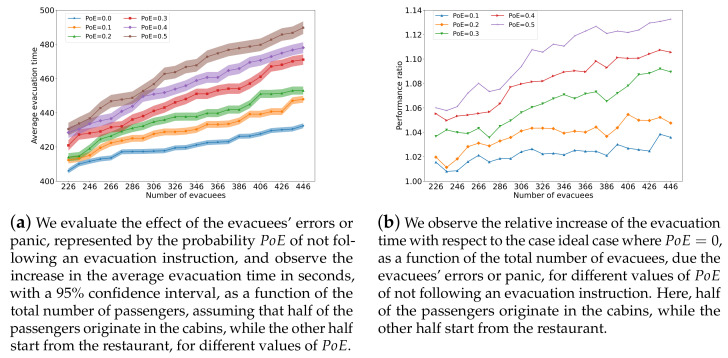
This figure evaluates the impact of the effect of errors or panic by the evacuees, as represented by the probability PoE. On the left, we show the increase in the average evacuation time in seconds, with a 95% confidence interval, as a function of the total number of passengers, assuming that half of the passengers originate in the cabins, while the other half start from the restaurant, for different values of PoE. Under the same conditions on the number of passengers starting from the cabins and the restaurant, the right-hand side figure shows the increase in the relative performance ratio in average evacuation time as compared to the ideal case with PoE=0 for different values of PoE and as a function of the total number of evacuees (passengers).

**Table 1 sensors-24-01850-t001:** Summary of the related work in this paper.

**Evacuation** **Review**	**Focus**	**Related work**
	Crowd monitoring	[4,6,25]
	Disaster detection & prediction	[3,6,25]
	Evacuation modelling	[6,11,15,19]
	Evacuation path planning	[6,11,15,28,30]
**Ship** **Accident**	**Accident type**	**Related work**
	Ship sinking	[12,13]
	Ship collision	[14]
**Evacuation** **Simulation**	**Simulation model**	**Related work**
	Agent-based model	[1,8,9,17,20,32]
	Social force model	[10,21,23,24]
	Social potential field model	[33]
	Flow model	[17,20]
**Path** **Planning**	**Planning method**	**Related work**
	A*	[5]
	Swarm optimization algorithm	[26]
	Proactiv & reactive method	[29]
	Cognitive packet network-based method	[7,38,39]
	OPEN	[31]
	Social potential field	[7]
	Temporally ordered routing algorithm	[34]
	Directional pathfinding method	[35]
	Table-driven method	[36]
	ANS	[37]
	Dinic algorithm	[40]
	Minimum spanning tree	[41]
	Hazard potential field	[42]
**Risk** **Analysis**	**Analysis method**	**Related work**
	Bayesian network	[16]
	Failure modes and effects analysis & analytic hierarchy process & fuzzy rule-based Bayesian reasoning & ER	[22]
**Person** **Localization**	**Method**	**Related work**
	Hybrid optimized fuzzy threshold extreme learning machine	[27]
**Decision Rule** **Learning**	**Method**	**Related work**
	Coevolutionary fuzzy rule miner	[2]
**Evacuation Analysis** **& Layout Optimization**	**Method**	**Related work**
	FDS+EVAC	[18]
**Wireless Energy** **Transmission**	**Method**	**Related work**
	Four-stage transmission	[43]
**Activity & Data** **Detection**	**Method**	**Related work**
	Multi-armed bandit	[44]

## Data Availability

The data presented in this study are available on request from the first author.

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
