# Peer review of "Impact of IoT System Imperfections and Passenger Errors on Cruise Ship Evacuation Delay"

_sensors, 2024, doi:10.3390/s24061850_

Round 1
Reviewer 1 Report
Comments and Suggestions for Authors
Recommendation
In this article "The scenario in terms of wireless or wired networks, the authors focused on especially in emergency situations when decisions and communications are frequently updated and many messages are sent to evacuees, and to the staff on the ship. This congestion can cause delays in updates regarding the navigation direction, and transmitting network packets and hence messages can be lost, and decisions may contain errors due to the use of delayed or obsolete data in the decision algorithms.
1- The authors must provide the full abbreviation for "ANS"
2- the author must improve the section 3 impact of delay based on solving the problem statement "cost components and discrete locations in a disaster-affected region"
3- When the disaster is approaching, later departures from evacuees may expose evacuees to severe environmental impacts, which may cause possible injuries or even loss of life. Worse, departing from evacuee. The authors must improve Section 3 and focus on reducing the risk assessment of the evacuation process and passenger evacuation capacity.
4- Please, give a good answer, What’s the difference between your work and the second article that you published on this website "arXiv:2306.14241v1 [cs.NI] 25 Jun 2023"
Comments on the Quality of English LanguageRecommendation
In this article "The scenario in terms of wireless or wired networks, the authors focused on especially in emergency situations when decisions and communications are frequently updated and many messages are sent to evacuees, and to the staff on the ship. This congestion can cause delays in updates regarding the navigation direction, and transmitting network packets and hence messages can be lost, and decisions may contain errors due to the use of delayed or obsolete data in the decision algorithms.
1- The authors must provide the full abbreviation for "ANS"
2- the author must improve the section 3 impact of delay based on solving the problem statement "cost components and discrete locations in a disaster-affected region"
3- When the disaster is approaching, later departures from evacuees may expose evacuees to severe environmental impacts, which may cause possible injuries or even loss of life. Worse, departing from evacuee. The authors must improve Section 3 and focus on reducing the risk assessment of the evacuation process and passenger evacuation capacity.
4- Please, give a good answer, What’s the difference between your work and the second article that you published on this website "arXiv:2306.14241v1 [cs.NI] 25 Jun 2023"
Author Response
The response to the reviewer is uploaded as a file.

Reviewer 2 Report
Comments and Suggestions for Authors
The article evaluates the influence of key features on the evacuation delay of passengers by means of communication network delays and the errors of evacuees during the emergency evacuation of the passengers from the ship. The results by using AnLogic simulator show the importance of the design of evacuation systems with realistic constraints and parameters in the ship environment. My comments are as below:
1. In line 55: 'this research' looks confusing. Need rewriting of the sentence or consider reorganizing para.
2. In related work, sufficient literature study related to emergency evacuation in ships or such equivalent disaster management scenarios needs to be added.
3. Full forms of all acronyms need to be included at appropriate places.
4. In Fig 2a/3a/4a, what is the Unit of average evacuation time? Is it seconds? or milliseconds?
5. As IoT is included in the title of the article, no discussion on IoT technologies, its processing delay, etc in such environments is discussed. Suggested references for IoT-based localization and disaster management are 10.1109/ICSCDS56580.2023.10105132, 10.3390/electronics12041051, 10.1016/j.eswa.2021.115500, 10.1109/ACCESS.2017.2752174, etc.
6. Please highlight the contributions of authors and novelty
7. Simulation result comparison with similar work would add much more value to the article.
8. Why is this scenario applicable for only ship evacuation? Why not others? Please include such discussion for better clarity.
9. The flow of the article is also to be relooked.
Comments on the Quality of English Language
Moderate updates required.
Author Response

(The authors gave the same response as above.)

Reviewer 3 Report
Comments and Suggestions for Authors
In general, the article is well done and can be recommended for publication in the journal “Sensors”. However, “Future Internet” and “IoT” journals can also be considered. This reviewer suggests accepting this article in the journal “Sensors” after increasing the emphasis on physical processes and the actual application of sensors.

Author Response

(The authors gave the same response as above.)

Round 2
Reviewer 1 Report
Comments and Suggestions for Authors
The recommendation
1-The objective of the manuscript must be precisely presented.
2- The authors should consider revising Figure 1 for clarity, perhaps by enhancing the visual representation and ensuring that common symbols are used for EMRs at the locations of the EDPs and other equations. This will help readers better understand the content without needing to decipher unfamiliar symbols in Eq. 1 etc.
3- Try to add a table in the Related Work section, summarizing all the work done.
Comments on the Quality of English LanguageThe recommendation
1-The objective of the manuscript must be precisely presented.
2- The authors should consider revising Figure 1 for clarity, perhaps by enhancing the visual representation and ensuring that common symbols are used for EMRs at the locations of the EDPs and other equations. This will help readers better understand the content without needing to decipher unfamiliar symbols in Eq. 1 etc.
3- Try to add a table in the Related Work section, summarizing all the work done.
Reviewer 2 Report
Comments and Suggestions for Authors
The author has reworked on the paper and substantially updated it. Now, it can be accepted in the present form.
Comments on the Quality of English LanguageThe author has reworked on the paper and substantially updated it. Now, it can be accepted in the present form.